# Influence of Residual Stress around Constituent Particles on Recrystallization and Grain Growth in Al-Mn-Based Alloy during Annealing

**DOI:** 10.3390/ma14071701

**Published:** 2021-03-30

**Authors:** Sung-Jin Park, Shinji Muraishi

**Affiliations:** Departments of Materials Science and Engineering, Tokyo Institute of Technology, Tokyo 152-8552, Japan; sjinpark4157@gmail.com

**Keywords:** recrystallization, plastic deformation, dislocations, inhomogeneity problem, residual stress

## Abstract

Effect of the residual stress on the recovery and recrystallization behaviors of the cold-rolled AA3003 aluminum alloy was investigated. The evolution of deformed microstructure and dislocation density characterized by TEM and Synchrotron X-ray measurements found that the change in the ratio between low angle grain boundaries (LAGBs) and high angle grain boundaries (HAGBs) during annealing is varied depending on the initial dislocation density, where the value of HAGB/LAGBs ratio is amounted to be about 0.8 at maximum. The nucleation and growth rate of the recrystallized grains are strongly dependent on the net driving pressure associated with dislocation density increased by the amount of reduction. EBSD analysis revealed that the deformed zone composed of the fine equi-axed grains with large misorientation angles would be formed in the vicinity of the constituent particles, which is consistent with the region of the large residual stress and total displacement predicted by Eshelby inhomogeneity problem under cold rolling condition.

## 1. Introduction

The optimum microstructure and mechanical properties of commercial AA3003 aluminum alloy are controlled by consecutive reduction and heat treatment, where concurrent precipitation would greatly influence the recovery and recrystallization at a certain temperature. It is well known that stored elastic energy by cold working process is the most important factor in the recrystallization of rolled aluminum alloys [1], however, recrystallization behavior is greatly influenced by concurrently existing particles and precipitates. Whether concurrent precipitation would occur or not during the homogenization and subsequent heat treatment, fine secondary particle causes a significant effect on recovery and recrystallization phenomena. Many researchers proved that promotion or suppression of concurrent precipitation occurred during annealing when correct homogenization conditions were selected [2]. Therefore, the initial microstructure introduced by homogenization treatment is an important factor for cold rolling and subsequent annealing treatment, where the volume fraction, size, shape, and inter-spacing of the particles can be varied to influence stored energy and recrystallization phenomena [3,4]. In our previous work [5], we reported the recrystallization of cold-rolled AA3003 aluminum alloy during annealing, where the change in electric conductivity associated with the concurrent precipitation as well as the apparent activation energy for recrystallization was discussed. It has been demonstrated that cold-rolled sheets homogenized by a certain temperature before cold rolling exhibited a negligible level of concurrent precipitation during annealing. Regarding the effect of secondary particles on recrystallization, many types of research have been carried out such as delay of the restoration behavior due to concurrent precipitation [6,7], and the promotion of recrystallization by particle-stimulated nucleation (PSN) [8,9]. 

Humphreys [10] have studied the influence of coarse precipitate on the recrystallization behavior of aluminum and its alloys during isothermal annealing, and the deformed structure around the coarse particles can be classified into two types: the distorted zones formed perpendicular to rolling direction and rotated zone formed parallel to rolling direction. These deformation zones formed by an accumulation of dislocation around the coarse particles are particularly important because they provide a PSN effect of recrystallization, which leads to a strong impact on recrystallized grain size and structure [1]. PSN is generally known to occur only in particles larger than about 1 μm in diameter. Since the probability of PSN is strain dependent, the resultant grain size as well as the texture orientation is possibly controlled by the distribution and the number of potential nucleation sites around the particle, in which the higher magnitude of the residual strain would be expected by the larger secondary particles.

The change of microstructure that occurs during annealing depends on the degree of the restoration mechanism such as recovery, recrystallization, and grain growth. The deformation zones formed around coarse primary particles (>1 μm) during plastic deformation may accelerate the recrystallization behavior by particle-stimulated nucleation (PSN) [11]. Whereas, the fine secondary particles suppress both the migration of low-angle grain boundary (LABG) and high-angle grain boundary (HABG) in Al-Mn-based alloy (Zener pinning) [12]. More importantly, the recovery and recrystallization kinetics are significantly retarded by large amounts of Mn supersaturation in Al-Mn-based alloy, which is commonly referred to as concurrent precipitation [13,14]. Therefore, in order to suppress the concurrent precipitation, microstructure controlling by homogenization treatment prior to the cold rolling and subsequent annealing is an important issue in advanced engineering processing in Al-Mn-based alloy [15,16].

The increase in the dislocation density during the cold rolling has the greatest influence on the driving force of the recrystallization and the velocity of boundary migration under subsequent annealing. Although the kinetics of the recrystallization accompanied with the concurrent precipitation during annealing depend on the activation energy of Mn diffusion in Al matrix [17,18], detail of the dislocation interaction with the constituent particles is still unclear, especially, the quantitative/macroscopic model for the evolution of dislocation microstructure associated with dislocation density during annealing is limited. Recently, the micromechanics-based analysis of the residual stress around the coarse particles under cold rolling condition has been proposed by the authors [19].

Therefore, we investigated the effect of the amount of reduction ratio and the subsequent annealing treatment on the recrystallization phenomena under the certain volume fraction of primary particle in AA3003 alloy, where the dislocation density and the misorientation of HAGBs and LAGBs were experimentally measured by synchrotron X-ray and EBSD analysis, respectively. Especially, the restoration of deformed microstructure around the constituent particles was discussed in terms of the residual stress around the α-Al(Mn, Fe)Si particle by solving an inhomogeneous inclusion problem under cold rolling condition.

## 2. Experimental Procedure

The chemical composition of AA3003 aluminum alloy in this study is listed in Table 1. The initial as-cast alloy (200 mm in length, 80 mm in width, 155 mm in thick) was fabricated by DC cast. Figure 1 schematically shows the dimension and orientation of DC-cast (Figure 1a) and cold-rolled (Figure 1c) AA3003 aluminum alloy. With the aim of reduction of concurrent precipitation during subsequent annealing, the homogenization treatment (as-homo) was conducted at 580 °C for 8 h with a slow heating rate of 50 °C/h, and then immediately quenched into the water at room temperature (R.T). Then the as-homo specimens were processed by cold-rolling with the amount of reduction, 20% (AM2), 50% (AM5), and 90% (AM9). In order to analyze the temperature dependency of the effect of concurrent precipitation on the recrystallization, the cold-rolled sheets were isothermally annealed at 300, 350, 375, and 400 °C for 2 s to 24 h in salt-bath, and then immediately quenched into the water at R.T.

The grain microstructure for the deformed and annealed specimens at the center of the thickness was characterized by an optical microscope (OM, BX51M, Olympus, Tokyo, Japan) and a field-emission scanning electron microscopy (FE-SEM, JSM7200F, JEOL Ltd., Tokyo, Japan), in which the longitudinal section is composed of the rolling direction (RD) and the normal direction (ND). The measurement of crystalline orientation and grain boundary misorientation was conducted by FE-SEM equipped with electron backscatter diffraction (EBSD, DVC5, EDAX, AMETEK Inc., Berwyn, PA, USA), in which the data was treated by TSL orientation imaging microscopy (OIM, Analysis8, EDAX, AMETEK Inc.) analysis software and MTEX toolbox for MATLAB (R2019a) [20]. The misorientation of grain boundaries of less than 2 degrees was neglected due to significant orientation noise in the deformed and annealed specimens. Generally, a low angle grain boundary (LAGB) is defined as boundaries with misorientation between 2–15 degrees, and boundaries exceeding 15 degrees are defined as high-angle grain boundary (HAGB) [21]. In order to examine dislocation microstructure, transmission electron microscopy (TEM) observation was carried out for as-rolled and annealed specimens with an operating voltage of 300 kV (JEM-3010, JEOL Ltd.). The preparations of TEM foils were made by twin jet polishing technique in a 20% HNO_3_ methanol solution under the condition of about −20 °C or less.

Electric conductivity was measured by eddy current inspection (using SIGMA tester), which is expressed in terms of %IACS (International Annealed Copper Standard). The microhardness was measured by Vickers hardness tester (MMT-X, Matsuzawa Co. Ltd., Akita, Japan) with an interval of 50 μm under a load of 0.98 N for 15 s. The change in dislocation density was measured and analyzed by synchrotron and conventional XRD. We utilized the high-intensity X-ray BL02B2 beamline (λ = 0.0516 nm) of the Spring-8 synchrotron radiation facility (JASRI, Hyogo, Japan) for this study. The time resolution of this experiment was 10 s. The conventional XRD measurement was carried out by convergent beam method with Cu-Kα (λ = 0.1540 nm) radiation (RINT-2100, Rigaku Co. Ltd., Tokyo, Japan). 

The residual stress field in/outside the precipitate embedded in the aluminum matrix was computed by analytical solutions of the Eshelby tensor [19,22,23,24], where the PSN effect on the deformation zone around the spherical inhomogeneity is assumed by mismatch of plastic strain between the α-Al(Mn, Fe)Si particle and Al matrix. The elastic properties used for the matrix and precipitate are given in Table 2. The displacement and von Mises equivalent stress field around the α-Al(Mn, Fe)Si precipitate are represented as density plots using MATLAB. 

## 3. Results

### 3.1. Microstructure of Homogenized Specimen

The typical microstructure and distribution of primary and secondary particles in the as-homo specimen are shown in Figure 2. The as-homo specimen shows characteristic equiaxed grain microstructure with average grain size of 160 μm (Figure 2a). In order to distinguish the primary particle Al_6_(MnFe) and α-Al(Mn, Fe)Si, the backscattered electron image is presented in Figure 2b. A large quantity of primary particles with rod-like, plate-like, and network eutectic shape are distributed in the grain boundaries and inside the inter-dendritic area (Figure 2a,b), where primary particles consist of α-Al(Mn, Fe)Si phases and less amount of Al_6_(Mn, Fe) phases with an average size of 2.5 μm. The Al_6_(Mn, Fe) phase was transformed to a large number of α-Al(Mn, Fe)Si phase, which can be clearly identified in the backscattered electron image (BEI) of the as-cast specimen and the homogenized specimens (Figure 2b), where the α-Al(Mn, Fe)Si phase exhibits brighter contrast than the Al_6_(Mn, Fe) phase. A large number of fine secondary particles were homogeneously formed in the Al matrix due to the effect of the homogenization treatment. It is noted that dislocation networks were frequently observed around the smaller precipitate with a diameter of 459 nm at the interior of the grain, which is enclosed by the dislocation walls (Figure 2c). The quantitative values of grain size, primary particle size, and electric conductivity are listed in Table 3. Since the electric conductivity of as-homo specimen was significantly increased as compared to that of as-cast specimen, the amount of concurrent precipitation during the recovery and recrystallization stages is potentially reduced by homogenization treatment.

### 3.2. Change in Microstructure with Increase of Strain

TEM images of cold-rolled specimens for AM2, AM5, and AM9 are shown in Figure 3. Relations between specimen orientation and viewing direction are indicated by TD, ND, and RD. It is apparent that, as the amount of reduction increases from AM2 to AM9, the cold-rolled specimens show elongated grain microstructure parallel to the RD. As the rolling reduction increases, the microstructure of the cold-rolled specimens shows elongated grains with the straight grain boundaries parallel to the rolling direction.

The ideal plastic strain of the cold rolling can be expressed by the change in the thickness of the cold rolled sheet as follows,
(1)$$\varepsilon_{t}=\int_{t_{0}}^{t_{1}}\frac{dt}{l}=\ln\left(\frac{t_{1}}{t_{0}}\right)=\ln\left(1+\varepsilon_{n}\right)$$
where the logarithmic strain (true strain) is *ε*_t_, the nominal strain is *ε*_n_, the reduced thickness of the cold rolled sheet is t_1_, and the initial thickness of the sheet is $t_{0}$.

It is seen from Figure 3a,c,e that thickness of the grains with HAGB and its crystallite size tend to decrease with increasing strain. The cell and sub-grain formed during the plastic deformation in AM2 and AM5 specimens are hardly changed (with the lower strain, *ε*_t_ ≤ 0.69), while the volume fraction of the grains with HAGB is increased in the AM9 specimen (Figure 3e). This implies that high-angle boundaries are newly developed by further plastic deformation, which leads to grain fragmentation due to the higher strain [27]. It is clearly seen from Figure 3b,d,f, which are the higher magnification images taken from ND; dislocation microstructure is composed of sub-grain boundaries (Figure 3b) and dislocation cells (Figure 3b,d). On the other hand, the strain contrast of AM9 is difficult to identify the boundaries clearly, which is due to the dense dislocations introduced by heavily plastic deformation (Figure 3f). 

In literature [28], the cell consists of a wide and diffused boundary containing dislocation tangle, while the sub-boundary is narrow, where the misorientation angle across the sub-boundary is larger than that across the cell wall. Accounting the bright field images, distinctive differences in crystal orientations are apparent, but the dense dislocations more influence the strain contrast at the grain interior for the higher magnitude of plastic strain in Figure 3f.

### 3.3. Deformation Zone around the α-Al(Mn, Fe)Si Precipitate with Increase of Strain

EBSD analysis was conducted for the deformed zones around the coarse α-Al(Mn, Fe)Si particles (>1 µm), and IPF (inverse pole figure) maps are represented in Figure 4, where black regions are α-Al(Mn, Fe)Si particles. The EBSD analysis was measured in a size of 5 µm × 10 µm with a step size of 0.04 µm. The respective IPF maps for the specimens with the different amounts of reduction, 20%, 50%, and 90%, demonstrate the evolution of the deformation zones with an increase of the plastic strain. In particular, elongated grain morphology along the direction parallel to RD is clearly distinguished by the color-coded plots, which indicate the extensive plastic flow around the undeformed α-Al(Mn, Fe)Si precipitate (Figure 4c). The deformation zone around the α-Al(Mn, Fe)Si precipitate introduced by cold rolling can be classified into two categories. The first is characterized by the distorted zone left and right of the coarse particle, and the other one is the rotated zone below and above the coarse particle. This rotated zone corresponds to the refined sub-grain structure and high lattice rotation, providing a high local stored energy, which is important for nucleation of recrystallization.

In order to distinguish the subgrain formation in the vicinity of the α-Al(Mn, Fe)Si precipitate, the local misorientation angle observed for the EBSD map of AM5 was replotted by the Kernel average misorientation (KAM) map and its line profile in Figure 5. The KAM map and the line profile of misorientation across the grain (black arrow) are obtained from the region indicated by the blue dotted line (region1 in Figure 4b). It is shown in Figure 5 that the misorientation profile is steeply increased to 9 degrees at a distance of 0.35 µm. After the stepwise increase and decrease of misorientation, there is no further change in the profile. This indicated that a well-developed sub-grain is formed inside the grain next to the coarse particle during plastic deformation.

Accounting for the non-uniformity of the misorientation angle in Figure 5b, the rotated zone consists of higher local misorientation angle, which reflects the existence of a subgrain boundary near the coarse precipitate (Figure 5a). This fact may be attributed to an exclusive lattice distortion induced at the matrix-precipitate boundary, which will be discussed in a later section of inhomogeneity problem.

In order to obtain the misorientation in the vicinity of the rotated zone quantitatively, line profiles were analyzed for AM2, AM5, and AM9 on the EBSD map (Figure 6). The misorientation profiles along RD and ND directions are taken from the regions indicated by dotted lines as shown in Figure 4. Black line shows the difference of the misorientation angle between analysis points, while the red line shows the accumulated value of misorientation. The misorientation angle around the precipitate depends strongly on the plastic strain, where the maximum angles of AM2, AM5, and AM9 amounted to 17.7, 31.8, and 61.2 degrees for RD directions (Figure 6a–c), and 29.6, 18.3, and 47.5 degrees for ND directions (Figure 6d–f), respectively. Moreover, numbers of the misorientation boundaries can be found in both the RD and ND directions for AM9 in Figure 6c,f.

Accounting for the misorientation profiles of the rotated zone formed around the coarse precipitate, it is suggested that the recrystallization of the cold-rolled sheet would be easily occur from the PSN effect under the large plastic deformation.

The effect of magnitude of plastic strain on the number fraction of the LAGBs and HAGBs are shown in Figure 7. The number fractions of LAGBs and HAGBs are indicated by the red and black lines, respectively. Note that the number fraction of LAGBs and HAGBs were evaluated by TSL OIM analysis software from the EBSD maps in the longitudinal section of the cold-rolled AA3003 aluminum alloy.

The number fraction of LAGBs drops with increasing the plastic strain, while the fraction of HAGBs increases in inverse proportion to the number fraction of LAGBs. Accounting for the fact that the profile of Vickers hardness as indicated by the blue line shows similar increasing profile as compared with the number fraction of HAGB, the fraction of HAGBs becomes sufficient to proceed with the recrystallization when the fraction of HAGBs reaches to 60~70%, which is a well-known fact pointed out by Humphrey et al. [29]. The number fraction of HAGBs in the cold-rolled specimens shows a parabolic increase with plastic strain, which is measured to be 7% for AM2, 57% for AM5, and 81% for AM9. The number fraction of HAGBs in the as-homo is measured to be 2.4%. Therefore, the recrystallization in the AM9 during annealing easily occurs with the large number fraction of HAGBs as compared with the AM2 and AM5 when the same annealing temperature is assumed.

### 3.4. Evolution of Dislocation Density with Increasing Strain

Figure 8 shows the effect of the amount of reduction on the (111) diffraction peaks measured by X-ray. The profile of the diffraction peaks in AM2 (logarithmic strain, *ε*_t_ = 0.22) is not significantly different from that of as-homo (*ε*_t_ = 0.0), but the diffraction angle is shifted slightly toward the lower angle. This shift of diffraction angle implies an increase of d-spacing in ND direction as a result of residual strain. The profile of AM5 (*ε*_t_ = 0.69) shows the broadening of diffraction peak at the similar peak position of AM2 (*ε*_t_ = 0.22). As the amount of reduction is increased, further broadening occurs, but the diffraction peak position is not significantly changed in AM9 (*ε*_t_ = 2.30). Accounting for the peak broadening with an increase of the amount of plastic deformation, inhomogeneous strain distribution and a small crystallite of a cold-rolled specimen are suggested.

The residual strain due to the cold rolling and subsequent annealing can be quantitatively analyzed by means of Williamson-Hall plots, in which the magnitude of lattice strain and the crystallite size can be separately determined from peak shift and width [30]. Diffraction profiles used for this analysis were (111), (200), (220), (311), and (222) reflections of the aluminum phase. According to Williamson-Hall equation, the effect of lattice strain, *ε*_L_, on the value of the full width at half maximum (FWHM) obtained from each peak can be mentioned as follows, [30].
(2)$$\Delta 2\theta \cos\theta =\left(0.9\lambda /D+2\varepsilon_{L}\sin\theta \right)$$
where Δ2*θ* is the FWHM of diffraction peak, *θ* is the Bragg diffraction angle, *λ* is the wavelength of X-ray, and *D* is the average crystallite size.

Figure 9a shows the FWHM values in the form of quadratic equations of the Williamson-Hall plot for each specimen. Note that, according to the first and second terms of the Williamson-Hall plot in Equation (2), the crystallite size and the lattice strain can be determined from the intercept and the slope of the linear regression profiles, respectively. Apparently, the intercept for 90% cold-rolled specimens in Figure 9a shows the largest value, which indicates the smallest crystallite size. Meanwhile, the slopes of the profiles are similar regardless of the amount of reduction. This implies that the average elastic strain is not significantly increased by further plastic strain.

The changes in crystallite size deduced from the XRD and TEM results are represented in Figure 9b. It is clearly seen that the crystallite size obtained by Equation (2) decreases with an increase of the amount of reduction, which shows good consistency with those obtained by TEM observation. In literature [31], since the crystallite size calculated from X-ray diffraction analysis by Equation (2) can be regarded as the average size of sub-grains and dislocation cells due to the coherently scattered X-rays, the dislocation cell size observed in AM2, AM5, and AM9 was used for the crystallite size deduced from TEM images, which amounted to 510 nm, 331 nm, and 214 nm, respectively.

The change in the dislocation density induced by cold rolling can also be determined from X-ray diffraction by assuming the lattice strain caused by dislocation lines. According to the relationship between the crystallite size and lattice strain obtained by Equation (2), the dislocation density, *ρ*, is given by Williamson and Smallman as follows [32], 

(3)$$\rho =\frac{{\left(3nK/F\right)}^{\frac{1}{2}}{\left(\varepsilon_{L}{}^{2}\right)}^{\frac{1}{2}}}{Db}$$
where *n* is the number of dislocations on each face of the matrix, *K* is constant depending on the strain distribution around the dislocation line, *F* is an interaction parameter, ${\left(\varepsilon_{L}^{}{}^{2}\right)}^{1/2}$ is root mean square strain, and b is Burgers vector. In literature [33], the constant of *K* assumed by the Cauchy strain distribution amounts to *K* = 25, whereas that by Gaussian strain distribution is nearly *K* = 4. In the absence of the extensive polygonization in dislocation arrangement, the dislocation density is calculated by assuming *n* ≈ *F*, and K = 4. Thus, Equation (3) can be simply given as

(4)$$\rho =\frac{2\sqrt{3}{\left(\varepsilon_{L}{}^{2}\right)}^{\frac{1}{2}}}{Db}$$

The effect of plastic deformation on the thickness of the grain for HAGB obtained by Equation (1) and TEM image analysis is represented in Figure 10. The thickness profiles are plotted by the black dashed line for XRD analysis and the black solid line for TEM image analysis as a function of the logarithmic strains, 0.22, 0.69 and 2.30 (corresponding nominal strains, 20%, 50% and 90%). It is noted that TEM foil normal to TD direction was taken from the center part of deformed specimens. It can be seen that the HAGBs thickness in ND direction is decreased rapidly with increase of plastic strain, which is similar to the crystallite size when the plastic strain is larger than *ε*_t_ = 1.5. This consistency might be due to the fact that the crystallite observed by TEM is mainly composed of the fine equi-axed grains (Figure 3). It should be mentioned that the thickness of HAGBs by TEM analysis is consistent with the theoretical prediction by Equation (1).

Regarding the change in the dislocation density as indicated by the blue dashed line in Figure 10, the dislocation density shows parabolic increase with respect to the plastic strain, which amounts to 1.66 × 10^14^ m^−2^ for AM2, 2.68 × 10^14^ m^−2^ for AM5 and 3.87 × 10^14^ m^−2^ for AM9. Accounting for the facts of the similar slopes for lattice strains in Figure 9a and the linear relationship between the crystallite size and the lattice strain in Equation (4), the change in dislocation density at the larger plastic strain is attributed to the change in crystallite size D as shown in Figure 9a and Figure 10.

### 3.5. Recrystallization Behavior in Cold Rolled Specimen with Different Reduction Ratio

The evolution microstructures of the AM2, AM5, and AM9 specimens annealed at 350 °C are shown in Figure 11. The AM2 specimen clearly shows inhomogeneous recrystallization after 300 s, where partially equi-axed grains have taken place in the vicinity of the intersection of deformed grains. By applying the longer annealing time over 600 s, discontinuous recrystallization and the uni-axial grain growth parallel to RD have been proceeded. By increasing the amount of reduction, the recrystallization microstructure of AM5 specimen shows homogeneous equi-axed grains within the shorter annealing time of 30 s. By further increasing the amount of reduction, the recrystallization in AM9 specimen is completed within a few seconds, where recrystallized grain size amounts to be 8 µm.

The change in the Vickers hardness and the dislocation density for AM2, AM5, and AM9 under the annealing temperature at 350 °C and 400 °C are represented in Figure 12. The profiles of dislocation density for AM2, AM5, and AM9 are indicated by symbols with square, circle and triangle, respectively. For comparisons, the profiles of the Vickers hardness are indicated by dotted lines (right hand side of ordinate). The dislocation density is computed by Equation (4), in which the crystal size and the average lattice strain in Equation (2) are used for the calculation. For comparisons, dislocation density of as-homogenized specimen is also plotted. It can be clearly seen that the profiles of the hardness and the dislocation density show good consistency, where the transition of the recovery to the recrystallization characterized by the rapid drop of the hardness profiles is well explained by the change in dislocation density. These results strongly suggest that the recrystallization behavior progresses with the dislocation absorption and relaxation of the lattice strains. When the recrystallization process is completed, the dislocation densities for the AM2, AM5, and AM9 specimens show similar values, which amount to be 0.71 × 10^14^ m^−2^, 0.87 × 10^14^ m^−2^ and 0.78 × 10^14^ m^−2^, respectively.

As it can be seen in Figure 11 and Figure 12, the recrystallization behavior is proceeded by a short period, especially the AM9 with large strain is completed within 5 s at 350 °C. In order to observe the evolution microstructure of the dislocations during the recrystallization more clearly, TEM observation was conducted for the specimen annealed at lower temperature of 300 °C. TEM bright field images for AM5 and AM9 annealed at 300 °C are shown in Figure 13 and Figure 14 respectively. As shown in Figure 13a, the pinning of dislocation by fine particle inside of the grain is observed for AM5 annealed for 120s. Most of dislocations were recovered in the upper and lower part of the fine particle, while the pinned dislocation remained due to the presence of the fine particle. After annealing for 600 s, a hexagonal-shaped dislocation network can be seen at the sub-boundary connected from region Ⅰ to Ⅱ (Figure 13b). The spacing of the dislocations in dislocation network changes depending on the distance from the high angle boundary. For instance, the mesh size at region-Ⅰ near the high angle grain boundary amounts to be 32 nm, while the spacing of the mesh at region-Ⅱ is 27 nm. Jones et al. [34] reported that the absorption of each interior dislocation by HAGB generally results in the formation and rearrangement of different boundary dislocations, and this series of dislocation absorption processes lead to the emission of arrayed dislocation in the grain boundary by stress relaxation from the boundary edge where the low angle boundary contacts the high angle boundary.

Judging from the transmission image of dislocation arrangement in Figure 13c, several boundaries consisting of the dislocation arrays are overlaid. For instance, the lath-shaped grain subdivided into two parts at the boundary as indicated by A-B-C is three-dimensionally connected to the edge of the HAGB as indicated by D and E. Furthermore, several dislocations isolated from the dislocation arrays can be seen in the bottom side of the lath-shaped grain. Accounting for the above observation results, the region enclosed by the dislocation arrays indicates the initial stage of the sub-grain formation, where dislocation boundary is gradually developed by the subsequent annealing due to the migration of the dislocation boundary.

Figure 14a–c shows a TEM microstructure of annealed AM9 specimen. For 5 s (Figure 14a,b), the recovery is delayed by the pinning effect due to the presence of a fine particle (white circle in Figure 14a), thereby the dislocation remaining around the fine particle inside the grain (Figure 14a). In the meanwhile, preferentially recrystallized grain without interior dislocation was frequently observed next to the coarse particle by PSN effect (Figure 14b). 

When annealing was conducted for the 60 s (Figure 14c), the recrystallized grain was formed with a convex curvature and a straight boundary pinned by fine particles (white circle). It is possible to predict the direction of growth of recrystallized grains of heavily deformed specimens that naturally depend on the migration characteristics of the high-angle boundary. Since the driving pressure increases as the radius of curvature increases, the grains with concave curvature are subjected to a greater driving pressure than convex curvature toward the center of curvature. Therefore, unless the boundary is formed in a straight line, the grain growth will occur on the convex curvature side. Further details on the driving pressure are discussed in a later section of thermodynamic pressure.

The recrystallized grains formed with convex curvature can be seen in Figure 14b,c, where a straight boundary pinned by particles is also observed. This implied that the recrystallized grain will grow in the direction of the convex curvature (indicated by white arrow) by further annealing, while the migration of straight boundaries will be retarded where the grain boundaries and dislocations are pinned by fine particles (white circle in Figure 14a,c).

The mechanism of subgrain growth in this specimen with low plastic deformation can be seen schematically in Figure 15. The cell structures enclosed by tangled dislocations are generated inside the grains by cold rolling (Figure 15a), which is changed into the subgrains by rearrangement of dislocations during the recovery stage (Figure 15b). By subsequent annealing, subgrain boundaries are coalesced with an adjacent subgrain, leading to a larger subgrain with an irregular shape of lower misorientation angle (Figure 15c). This continues until the subgrains grow to a size of grain surrounded by HAGB during recrystallization, from which point the grain will begin to grow and abnormal grain growth can occur.

## 4. Discussions

### 4.1. Evolution of Dislocation Density

The dislocation density tends to increase in proportion to plastic strain, which leads to the formation of LAGBs by dislocation rearrangement. In principle, the recovery process is considered as the evolution of sub-grains, which is associated with the migration of LAGBs at the region with the large strain energy differences or orientation gradients [35]. 

The evolution of the dislocation density and crystallite size for AM2, AM5, and AM9 under the annealing temperature at 350 °C and 400 °C are represented in Figure 16. The profiles of dislocation density and crystallite size for AM2, AM5, and AM9 are represented by solid and dotted lines, respectively, where annealing temperatures are indicated by symbols with a black square for 350 °C and red triangle for 400 °C. The respective recovery and recrystallization stages during annealing were indicated as stage Ⅰ and Ⅱ, where the distinction between both of the stages as a function of annealing time was obtained via the modified Johnson-Mehl-Avrami-Kolmogorov (JMAK) microhardness model in the previous study [5].

As shown in Figure 16, the coarsening of crystallite occurred faster at 400 °C than low temperature at 350 °C, thereby the size of new recrystallized grains decreased in inverse proportion to annealing temperature. Therefore, a large number of nuclei for recrystallization are created by rapid migration and consumption of dislocations at high temperatures, leading to refinement of the grains by impingements between the new recrystallized grains.

The relationship between crystallites size and dislocation density during annealing could be considered in terms of the dislocation annihilation, nucleation, and growth for recrystallization. Since the stored energy of the formed subgrain structure during recovery is still large, it would be further lowered by the coarsening of the subgrain during recrystallization. It can be seen in Figure 16a, AM2 exhibits little change in the dislocation density and crystallite size through a relatively slow stage-Ⅰ at 350 °C, and rapid coarsening occurred at the beginning of stage-Ⅱ. 

Accounting for the fact that the dislocation density is sharply dropped at stage-Ⅱ, energy dissipation by dislocation annihilation under the stage-Ⅰ of AM2 is not significant. This indicates that the larger thermal activation energy is necessary for the migration of LAGB since the mobility of LAGB is essentially smaller than that of HAGB.

### 4.2. Driving Force for Recrystallization

The activation energy for migration of the low angle boundary in aluminum alloys depends on the activation energy for lattice-diffusion of the main additive element (Mn = 217 kJ/mol) [36], while the activation energy for migration of HAGBs in the aluminum alloy is close to the value for grain boundary diffusion (84 kJ/mol) reported by R. W. Balluffi [37]. In this study, it can be expected that the presence of alloying elements such as Mn, Si, and Fe will influence the increase of second-phase particles, which can significantly exert the effect in the pinning of grain boundaries and retarding the migration of grain boundaries. Significant pinning effect on the grain recovery and recrystallization was found by homogenized specimens, which is attributed to the formation of fine precipitates on both low and high angle grain boundaries. In our previous study [5], the apparent activation energies for recrystallization of AM2, AM5, and AM9 were determined to be 332 kJ/mol, 239 kJ/mol, and 115 kJ/mol, respectively, where the apparent activation energy for recrystallization of AM9 is almost similar to the activation energy for self-diffusion in the aluminum. Therefore, recrystallization kinetics for AM9 with an increase of the amount of reduction is attributed to the stored energy associated with dislocation density, which increases the nucleation sites and the grain boundary mobility. 

The changes in the number fraction of LAGBs and HAGBs associated with grain boundary energies *γ*_b_ under cold rolling conditions are important when discussing the mobility of grain boundaries during annealing. The energy of simple tilt boundary is given by Read-Shockley [38].
(5)$$\gamma_{b}=\gamma_{\max}\frac{\theta }{\theta_{\max}}\left(1-\ln\frac{\theta }{\theta_{\max}}\right),\;\left(0<\theta \;\leq \;\theta_{\max}\right)$$
where *θ*_max_ is high angle boundary (commonly considered 15 degrees) and *γ*_max_ is the value of boundary energy when the boundary is at a high angle. *γ*_max_ is considered to be 0.324 J/m^2^ in the aluminum [39]. 

According to Equation (5), the energy of the low angle boundary increases with increasing misorientation *θ*, while the energy per dislocation decreases with increasing misorientation. In view of geometrically necessary dislocations [40], the contribution of the dislocation self-energy would be negligibly smaller than that of dislocation interaction energy when a certain boundary is assumed by the large numbers of dislocations. 

Assuming that the energy stored by plastic deformation is dissipated by recrystallization and grain boundary migration when the towing possibility, dislocation annihilation, or dislocation absorption into the boundary are accompanied, the stored energy, E_d_, can be estimated from the dislocation density in view of self-energy of dislocations.
(6)$$E_{d}=c_{2}\rho {\mathrm{Gb}}^{2}$$
where, c_2_ is the pre-factor for dislocation energy (0.5 is often used for self-energy of dislocation by line tension approximation). Substituting dislocation densities calculated in the present study, 1.66 × 10^14^ m^−2^ for AM2, 2.68 × 10^14^ m^−2^ for AM5 and 3.87 × 10^14^ m^−2^ for AM9, and material constants, G = 2.6 × 10^10^ N/m^2^, b = 0.286 nm, stored energy for AM2, AM5, and AM9 amounts to 3.1 × 10^5^, 4.9 × 10^5^ and 7.1 × 10^5^ J/m^3^, respectively. Accounting for the equilibrium of thermodynamic pressure, critical radius of recrystallized grain can be predicted by the following formula,
(7)$$\gamma_{crit}=2\gamma_{b}/E_{d}$$
where *γ*_b_ is the grain boundary energy for 0.324 J/m^2^ in the aluminum, and *E*_d_ is the stored energy (thermodynamic pressure).

By substituting thermodynamic pressure (stored energy) and grain boundary energy, the critical radius of recrystallized grain yields approximately 1.8 μm for AM2, 1.3 μm for AM5, and 0.9 μm for AM9. This result indicates that AM9 shows good consistency in prediction and observation results due to higher driving force for recrystallization (Figure 15b,c), which also indicates that the region of the recrystallization would not be restricted around the fine secondary particles during annealing. meanwhile, the values of critical radius predicted for AM2 and AM5 specimens are deviated from the observation results, which is reflected by the larger apparent activation energy for AM2 (331 kJ/mol) and AM5(239 kJ/mol). Although the activation energy for recrystallization in the AM5 specimen is similar to the activation energy for Mn diffusion (217 kJ/mol) [36] in aluminum, since the concurrent precipitation did not occur during recovery and recrystallization in the present study, the activation energy for recrystallization of AM2 and AM5 may be influenced by the pinning effect of fine secondary particles. In particular, accounting for the fact that crystallite size of AM2 hardly changes during recovery, driving pressure for recrystallization and migration might be insufficient against the considerable pinning effect of fine particles on LAGB and HAGB. Hence, the fine secondary particle on LAGB and HAGB is responsible for the significant delay in recovery and recrystallization of AM2 and AM5.

### 4.3. Thermodynamic Pressure for Boundary Migration

The initial thickness of grains are gradually decreased to several hundreds of nm depending on the plastic strain (Figure 3e). Such a decrease in thickness of high angle grain leads to the large number fraction of HAGBs. In addition, the constituent particles accompanied with the local plastic deformation of the matrix phase may act as an important factor to promote recrystallization.

In the case of lower plastic deformation (AM2 and AM5), considerable time is required for recovery and relaxation of dislocation microstructure, which is attributed to a lower driving force associated with dislocation density for recovery. Although the recovery stage for AM5 and AM9 in the present study was completed in a very short time, the stored energy for AM2 in Figure 16 is calculated by using Equation (6), which amounted to 3.1 × 10^5^ J/m^3^ (the as-rolled), 2.8 × 10^5^ J/m^3^ (annealed at 350 °C for 2 min), and 2.7 × 10^5^ J/m^3^ (annealed at 400 °C for 2 s). Hence, the energy consumption at 350 °C and 400 °C during the recovery stage amounted to 8.1% and 11.1%, respectively. These results are consistent with the consumed energy as reported in [41], where 10% of the stored energy is consumed in the recovery process. 

In literature [1], the tangled dislocations are changed into the dislocation wall, which is further developed into three-dimensional dislocation cell and sub-grains with increase of the plastic strain. Furthermore, thermal activation by the heat-treatment process promotes the dislocation rearrangement and boundary migration. It can be seen in Figure 13c that the number of dislocations inside the grain is decreased by annealing at 300 °C for 600 s, which leads to the formation of a well-developed subgrain with a size of about 1 μm in the recovery stage. It is noted that, although the OM microstructure is not represented, recrystallization is not confirmed under the same annealing condition of AM5 (at 300 °C for 600 s). Therefore, the transition of the dislocation microstructure from the cell wall to the subgrain boundary is supposed to be the representative behavior in the recovery process. Hence, the rearrangement of dislocations for the formation of sub-grains is considered to be dominant rather than the annihilation of dislocations in the recovery process.

According to the literature [1], the net driving pressure *P*_net_ that acted on the boundary of recrystallized grain is given as,
(8)$$P_{\mathrm{net}}=P_{d}-P_{p}-P_{c}=\frac{\alpha \rho {\mathrm{Gb}}^{2}}{2}-\frac{3\rho f_{v}\gamma_{b}}{d}-\frac{2\gamma_{b}}{r}$$
where the pressures associated with dislocation density, pinning effect by particles, and boundary curvature are *P*_d_, *P*_p_ and *P*_c_, respectively. α is a constant of the order of 0.5, G is the shear modulus, *f*_v_ is the volume fraction of second-phase particle, *γ*_b_ is the boundary energy, and d is the diameter of particle. Recalling the results of Section 3.2 and Section 3.4, it can be seen that the net driving pressure associated with dislocation density and curvature of boundary affecting the nucleation and growth rate of recrystallized grains in AM9 is greater than AM2 and AM5 with lower plastic deformation.

Regarding the higher magnitude of plastic strain (AM9), the large numbers of smaller recrystallized grains would be formed with the rapid recovery and nucleation rate. According to the boundary energy in Equation (2), since the large magnitude of plastic strain induced the recrystallized grain with smaller radius of curvature (Figure 15), the retarding pressure for the boundary migration would be increased as suggested in Equation (8). Hence, the rapid recovery and nucleation rate are merely due to the higher stored energy, which exceeded the retarding pressure for the boundary migration. Furthermore, as can be seen in Figure 15c, the growth and migration would be delayed due to the pinning effect of particles. This indicates that, although the recrystallized grain would be firstly introduced at the rotation zone around the coarse particle, the region of the recrystallization is extended into the grain interior because of smaller retarding pressure. Eventually, the recrystallized grains are surrounded by the coarse particles.

This can be supported by the results analyzed by the EBSD and FE-SEM. The recrystallized grains are preferentially generated next to the coarse precipitate (white region, in Figure 17a), where the grain orientation spread (GOS) of aluminum indicates misorientation of less than 3 degrees [42]. The convex curvature of recrystallized grain reflects the pinning effect of fine particles as can be seen in Figure 17b.

### 4.4. Residual Stress of α-Al(Mn, Fe)Si Particle

Since the contribution of second-phase particles on the work hardening is significant, the dislocation microstructure strongly depends on the distribution microstructure of the second-phase under a certain amount of reduction. This indicates that the magnitude and gradient of plastic strain are locally changed with the spatial distribution of the particles, which is commonly referred to as the particle deformation zone [43]. This deformation zone provides a strong driving force for the nucleation at the local point during the initial recrystallization process. Accounting for the fact that the deformation zones formed by an accumulation of dislocation around the second-phase particles are introduced by a mismatch of plastic strain between the Al matrix and the inhomogeneous precipitate, residual strain and stress around the second particle can be treated by the inhomogeneous inclusion problem under cold-rolling condition [19,22].

According to the Eshelby theory, the stress disturbed by the inhomogeneity can be mentioned by assuming mismatch of the plastic strain, $\varepsilon_{\mathrm{ij}}^{p}$, as follows,
(9)$$\sigma_{\mathrm{ij}}=C_{\mathrm{ijkl}}^{*}\left(\varepsilon_{\mathrm{kl}}^{0}+S_{\mathrm{klmn}}\varepsilon_{\mathrm{mn}}^{**}-\Delta \varepsilon_{\mathrm{kl}}^{p}\right)$$
(10)$$\Delta \sigma_{\mathrm{ij}}=C_{\mathrm{ijkl}}^{*}\left(S_{\mathrm{klmn}}\varepsilon_{\mathrm{mn}}^{**}-\Delta \varepsilon_{\mathrm{kl}}^{p}\right)=C_{\mathrm{ijkl}}\left(S_{\mathrm{klmn}}\varepsilon_{\mathrm{mn}}^{**}-\varepsilon_{\mathrm{kl}}^{**}\right)$$
with
(11)$$\varepsilon_{\mathrm{mn}}^{**}=\varepsilon_{\mathrm{mn}}^{*}+\Delta \varepsilon_{\mathrm{mn}}^{p}$$
where $\varepsilon_{\mathrm{mn}}^{**}$ is total eigenstrain, $\Delta \varepsilon_{\mathrm{mn}}^{p}$ is the mismatch of the plastic strain, $\varepsilon_{\mathrm{mn}}^{*}$ is equivalent (fictitious) eigenstrain, and $C_{\mathrm{ijkl}}$ and $C_{\mathrm{ijkl}}^{*}$ are the stiffness of the matrix and the precipitate, respectively. Note that in the absence of the mismatch of the plastic strain in Equation (9) ($\Delta \varepsilon_{\mathrm{mn}}^{p}$ = 0), the relation of the total eigenstrain and the applied strain becomes
(12)$$\left(S_{\mathrm{klmn}}+{\left(C_{\mathrm{klij}}^{*}-C_{\mathrm{klij}}\right)}^{-1}C_{\mathrm{ijmn}}\right)\varepsilon_{\mathrm{mn}}^{**}=\varepsilon_{\mathrm{pq}}^{0}$$

The residual stress, $\Delta \sigma_{\mathrm{ij}}$, can be simply given by substituting the applied strain, $\varepsilon_{\mathrm{ij}}^{0}=0$ [19,22],
(13)$$\Delta \sigma_{\mathrm{ij}}=C_{\mathrm{ijkl}}^{*}\left(S_{\mathrm{klmn}}\varepsilon_{\mathrm{mn}}^{**}-\Delta \varepsilon_{\mathrm{kl}}^{p}\right)=C_{\mathrm{ijkl}}\left(S_{\mathrm{klmn}}\varepsilon_{\mathrm{mn}}^{**}-\varepsilon_{\mathrm{kl}}^{**}\right)$$

Thus, the relation of the total eigenstrain and the mismatch of the plastic strain is given as [19,22],
(14)$$\varepsilon_{\mathrm{mn}}^{*}={\left(\left(C_{\mathrm{ijkl}}^{*}-C_{\mathrm{ijkl}}\right)S_{\mathrm{klmn}}+C_{\mathrm{ijkl}}I_{\mathrm{klmn}}\right)}^{-1}C_{\mathrm{ijkl}}^{*}\Delta \varepsilon_{\mathrm{kl}}^{p}$$

Once the eigenstrain is solved, the stress inside the inhomogeneity can be computed by Equations (9) and (10). The residual stress outside the α-Al(Mn, Fe)Si precipitate can be derived from the exterior point Eshelby tensor as follows,
(15)$$\sigma_{\mathrm{ij}}\left(\mathrm{outside}\right)=C_{\mathrm{ijkl}}{\bar{G}}_{\mathrm{klmn}}\varepsilon_{\mathrm{mn}}^{**}$$
where ${\bar{G}}_{\mathrm{klmn}}$ is the exterior point Eshelby tensor. The explicit expressions of the exterior Eshelby tensors $\bar{G}$ for ellipsoidal inclusions are given by Ju and Sun [24]. Full details of all the mathematical basis associated with micromechanics are given elsewhere [19,22,23,24,40,44].

Note that, under the assumption that the plastic deformation is dominant in the aluminum matrix, the mismatch of the plastic strain in Equation (9) can be set as
$\Delta \varepsilon_{\mathrm{kl}}^{p}$ = $-\varepsilon_{kl}^{0}$. In order to predict the shape of the deformation zone, the distribution of the residual stress around the α-Al(Mn, Fe)Si precipitate is evaluated by the von Mises equivalent stress, $\sigma_{\mathrm{eqv}}$, with the following formula,


(16)$$\sigma_{\mathrm{eqv}}=\sqrt{\frac{1}{2}\left[{\left(\sigma_{11}-\sigma_{22}\right)}^{2}+{\left(\sigma_{22}-\sigma_{33}\right)}^{2}+{\left(\sigma_{33}-\sigma_{11}\right)}^{2}+6{\left(\sigma_{12}^{2}+\sigma_{23}^{2}+\sigma_{31}^{2}\right)}^{2}\right]}$$


Substituting the nominal plastic strain of the present study, $\Delta \varepsilon_{33}^{p}$ = $-\Delta \varepsilon_{11}^{p}$ = 0.2 (AM2), 0.5 (AM5) and 0.9 (AM9), and $\Delta \varepsilon_{22}^{p}$ = 0 into Equation (9), the total eigenstrain is amounts to $\varepsilon_{33}^{**}$ = $-\varepsilon_{11}^{**}$ = 0.299, $\varepsilon_{22}^{**}$ = 0 for AM2, $\varepsilon_{33}^{**}$ = $-\varepsilon_{11}^{**}$ = 0.747, $\varepsilon_{22}^{**}$ = 0 for AM5 and $\varepsilon_{33}^{**}$ = $-\varepsilon_{11}^{**}$ =1.344, $\varepsilon_{22}^{**}$ = 0 for AM9. The residual stresses and the von Mises equivalent stress computed by Equations (9) and (16) are listed in Table 4. It should be noted that the computed values of stress in Table 4 are greatly large in magnitude, even if the mismatch of plastic strain under the cold rolling condition is assumed to be $\Delta \varepsilon_{33}^{p}$ = $-\Delta \varepsilon_{11}^{p}$ = 0.2 (the amount of reduction, 20%).

Note that the magnitude of the residual stress is proportional to that of eigenstrain strain within the context of linear elasticity. Accounting for the stress level of plastic deformation as a yield criterion, the spatial distribution of the stress outside the inhomogeneity is important to consider the stress accumulation and relaxation phenomena under cold rolling. Assuming that the matrix phase is exclusively deformed, while the particle is completely elastic under the cold rolling, the spatial distribution of the plastic deformation is reproduced by summing the uniform plastic strain and the disturbed plastic strain in the matrix phase. Accounting for this fact, it is apparent that the residual stress field would be canceled by the opposite sign of uniform eigenstrain in inhomogeneity. Moreover, in view of average theory, this stress relaxation can be attained by the eigenstrain assumed outside the inhomogeneity, which is more suited for the stress relaxation by dislocations. Note that the magnitude of the disturbed plastic strain is inversely proportional to the considering volume of the region enclosing the inhomogeneity in equivalent inclusion system though Equation (10). This concept of stress relaxation is based on the Tanaka-Mori theorem [22,45,46] and the dislocation punching model reported by Shibata et al. [47]. 

Therefore, in order to realize the displacement field around the inhomogeneity, total displacement inside/outside the inhomogeneity is computed by superposing the elastic and plastic displacements. Recalling the fact that the equilibrium shape of the inhomogeneity given from Equations (9) and (10) is that without the uniform plastic strain in the solid, superposition of the uniform plastic strain can realize the deformed shape of the inhomogeneity in the matrix phase. Note that the displacement solutions for interior and exterior points are solved analytically by using following expressions [22,48,49],
(17)$$u_{i}\left(x\right)=1/8\pi \left(1-\nu \right)\left(\varepsilon_{\mathrm{jl}}^{*}\psi_{\mathrm{jli}}-2\nu \varepsilon_{\mathrm{mn}}^{*}\phi_{i}-4\left(1-\nu \right)\varepsilon_{\mathrm{il}}^{*}\phi_{l}\right)$$
with

(18)$$\phi_{i}=-x_{i}I_{I}\left(\lambda \right)$$(19)$$\psi_{\mathrm{ijl}}=-\delta_{\mathrm{ij}}x_{l}\left(I_{L}\left(\lambda \right)-\alpha_{I}^{2}I_{\mathrm{IL}}\left(\lambda \right)\right)-x_{i}x_{j}{\left(I_{J}\left(\lambda \right)-\alpha_{I}^{2}I_{\mathrm{IJ}}\left(\lambda \right)\right)}_{l}-\left(\delta_{\mathrm{il}}x_{j}+\delta_{\mathrm{jl}}x_{i}\right)\left(I_{J}\left(\lambda \right)-\alpha_{I}^{2}I_{\mathrm{IJ}}\left(\lambda \right)\right)$$
where I-integrals for spherical inhomogeneity were used for the calculation [22]. Total eigenstrain given from Equation (14) is substituted into Equation (17).

Computed results of the total displacement and the residual stress solved by Equations (17) and (9) are represented in Figure 18. It can be seen from the deformed shape, the total displacement of the matrix phase is strongly constrained by the inhomogeneity, where the thickness of the region at the right and left part of inhomogeneity are hardly changed in ND direction. Moreover, judging from the signs of $\sigma_{11}$ and $\sigma_{33}$, dilatational stress can be found in the vicinity of the precipitate along the $x_{1}$-axis (red, positive) and $x_{3}$-axis (blue, negative). Therefore, in order to cancel and relax the large magnitude of residual stress, multiple slip system of crystal dislocations should be operated, in which the volume average of the dislocation swept area should be comparable to the equivalent eigenstrain assumed in the inhomogeneity as predicted in Equation (14).

Accounting for the spatial distribution of the residual stress, it is natural to consider that the misorientation of the deformed zone is accompanied with the fine equi-axed grains surrounding the inhomogeneity as shown in Figure 4c. In fact, this can be supported by EBSD analysis (Section 3.3) for the deformed microstructure and misorientation angle at the deformed zone parallel to the rolling direction near the particles. The profile of misorientation angles predicted by the displacement gradient and measured by EBSD analysis is represented in Figure 19. The misorientation angles of the prediction and measurement results are indicated by the dotted and solid lines, respectively. The maximum misorientation angle at the rotated zone near the precipitate as indicated by the red solid line amounted to be 45 degrees at a distance of 1.9 μm from the interface for AM5 (misorientation measured by point to origin in Figure 6b), which is similar to the misorientation angle 45 degrees calculated from the gradient of the displacement at the same position (red dotted line in Figure 19).

## 5. Conclusions

The deformation and recrystallization of cold-rolled AA3003 aluminum alloy were investigated in terms of the dislocation density, the misorientation angle, and residual stress around the constituent particles. The dislocation density measured by X-ray was proportionally changed with the amount of reduction of the cold-rolled sheets, which shows good consistency with the TEM observation results and microhardness profiles. In the meanwhile, the constant value of lattice strain was found in X-ray strain analysis regardless of the amount of plastic strain. These results suggest that the HAGBs would be developed by dislocation absorption into the boundaries by further plastic deformation, which eventually leads to an increase in the driving force for the recrystallization. By subsequent annealing, the further development of the HAGBs was observed especially during the recrystallization, where the microhardness and the dislocation density are sharply decreased with an increase of the annealing time. This implies that the dislocation annihilation and dislocation absorption into the boundary occurred exclusively during the recrystallization. 

The residual stress analysis based on the Micromechanics theory revealed that the region of the residual stress resembles the deformed zone around the coarse particle, which is attributed to the fact that the elastic and plastic deformation of the matrix phase are strongly constrained by inhomogeneity. The prediction results of the displacement gradient along the rolling direction are consistent with the line profiles of the misorientation angle measured by EBSD analysis. These results suggest that the deformed zone composed of the fine equi-axed grains would be formed under the relaxation process of the residual stress due to local plastic deformation in the vicinity of the coarse particle, which may be responsible for the PSN effect on the recrystallization.

## Figures and Tables

**Figure 1 materials-14-01701-f001:**
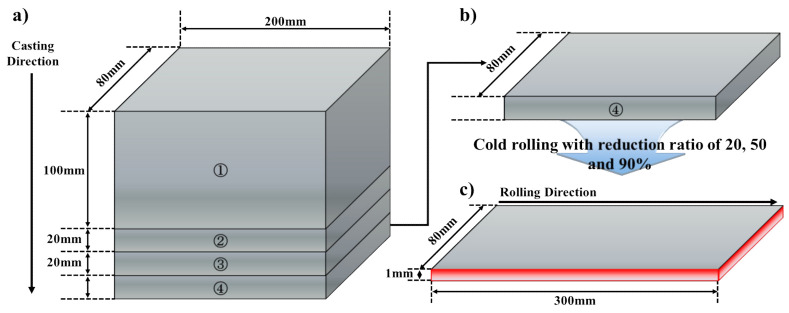
(**a**) Schematic diagram of DC casting direction and dimensions, (**b**) positions of a sample taken from the ingots for a test, and (**c**) dimensions of cold-rolled sheets.

**Figure 2 materials-14-01701-f002:**
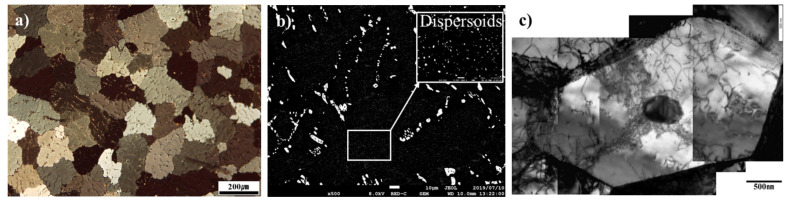
The typical microstructure of homogenized AA3003 aluminum alloy observed by (**a**) OM, (**b**) SEM-BEI, (**c**) TEM.

**Figure 3 materials-14-01701-f003:**
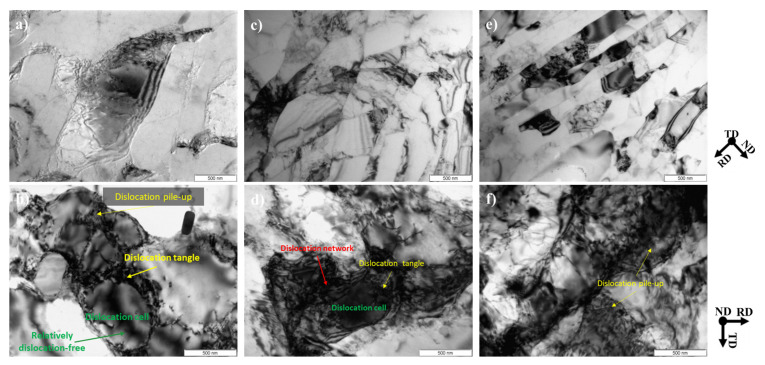
Representative TEM bright images of cold-rolled AA3003 aluminum alloy with different amounts of reduction. (**a**,**b**) 20%, (**c**,**d**) 50%, and (**e**,**f**) 90%.

**Figure 4 materials-14-01701-f004:**
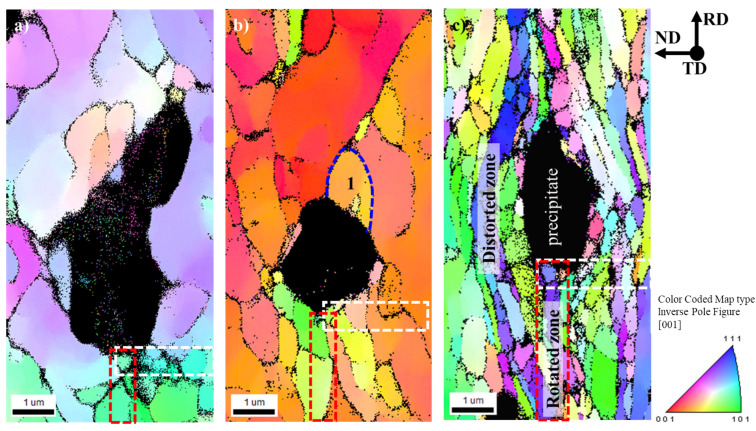
EBSD map around the α-Al(Mn, Fe)Si precipitate in the longitudinal section of the cold-rolled AA3003 aluminum alloy with different amounts of reduction, (**a**) 20%, (**b**) 50%, and (**c**) 90%. The inverse pole figures (IPF) indicate the color key of the crystal direction parallel to the normal direction (ND).

**Figure 5 materials-14-01701-f005:**
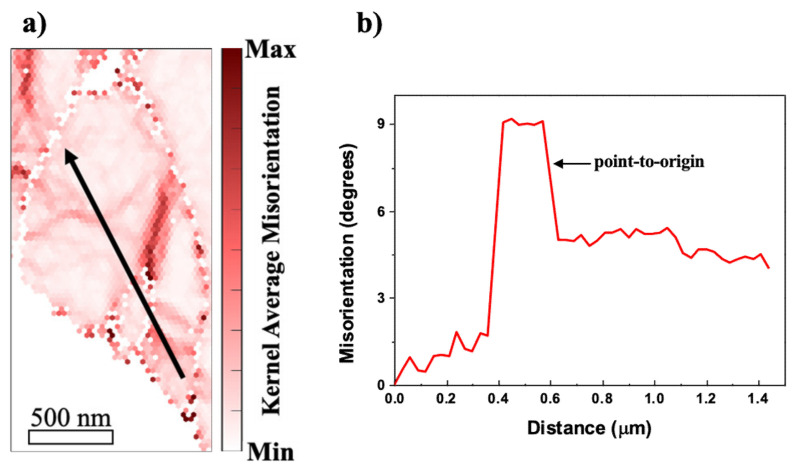
(**a**) Kernel average misorientation map of AM5 indicated in Figure 4b (region 1) with the arrow marking (**b**) misorientation profile.

**Figure 6 materials-14-01701-f006:**
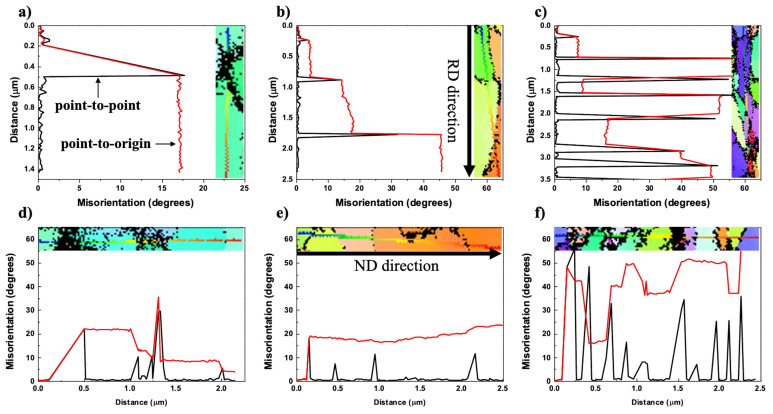
Misorientation angle profiles of (**a**,**d**) AM2, (**b**,**e**) AM5, and (**c**,**f**) AM9 across the rotated zone formed around α-Al(Mn, Fe)Si precipitate. The point-to-point line (black) indicates the profile of the orientation changes between adjacent point. The point-to-origin line (red) represents the profile of the orientation changes between all points and origin point.

**Figure 7 materials-14-01701-f007:**
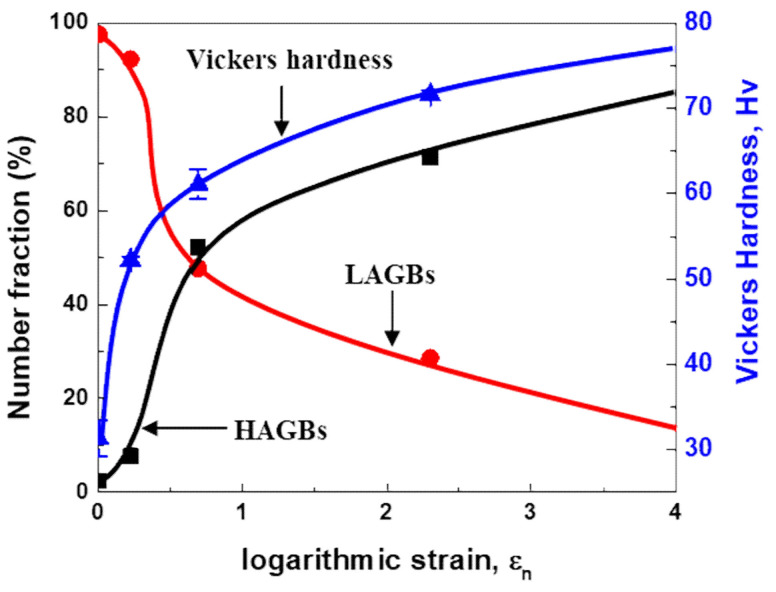
Changes in the number fraction of the LAGBs and HAGBs, and Vickers hardness with increase of the amount of reduction in the AA3003 aluminum.

**Figure 8 materials-14-01701-f008:**
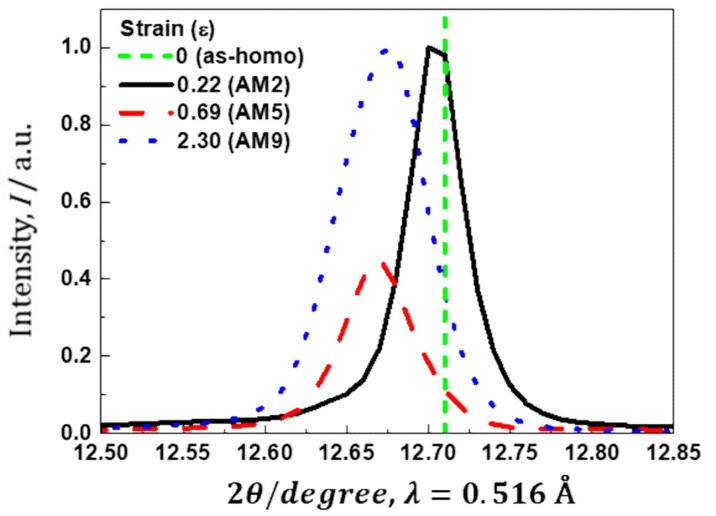
(111) diffracted peak at different amounts of reduction for the DC cast AA3003 aluminum alloy with homogenization.

**Figure 9 materials-14-01701-f009:**
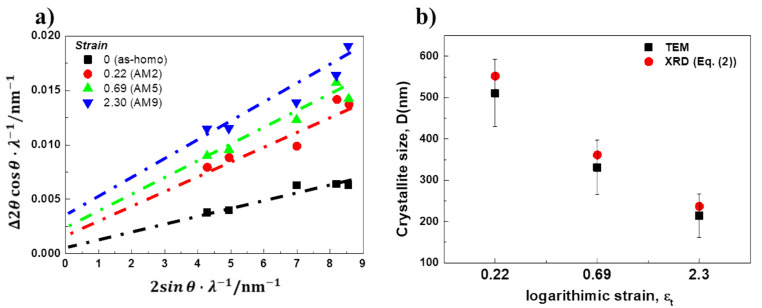
(**a**) Williamson-Hall plots and (**b**) Crystallite size of the cold-rolled AA3003 aluminum alloy measured by XRD and TEM.

**Figure 10 materials-14-01701-f010:**
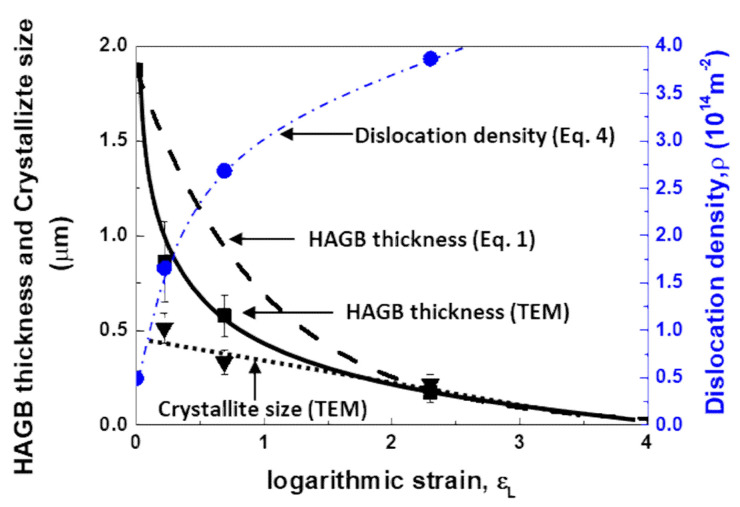
Evolution of the HAGB thickness, crystallite size, and dislocation density with increasing the logarithmic strain in the AA3003 aluminum alloy.

**Figure 11 materials-14-01701-f011:**
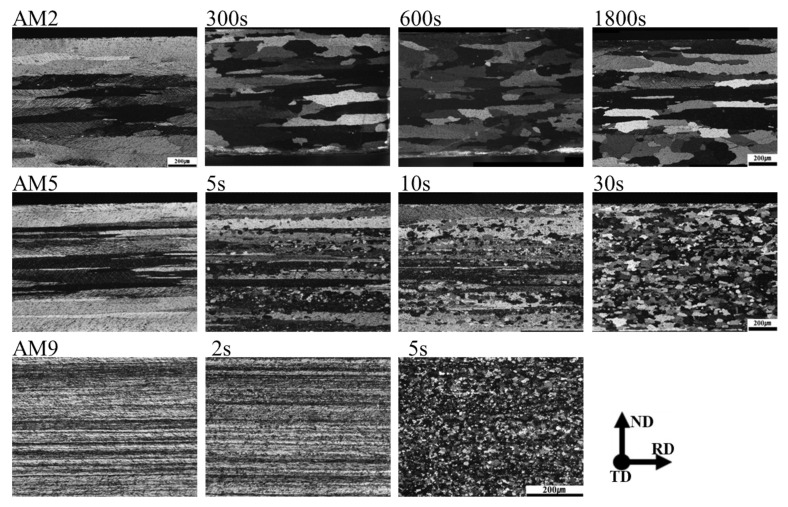
Evolution of the recrystallization microstructure for the AM2, AM5, and AM9 at 350 °C for different holding times.

**Figure 12 materials-14-01701-f012:**
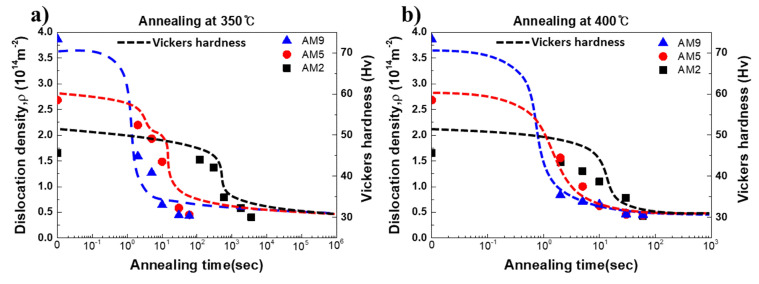
Change in the dislocation density as a function of annealing time at (**a**) 350 °C and (**b**) 400 °C in AM2, AM5, and AM9.

**Figure 13 materials-14-01701-f013:**
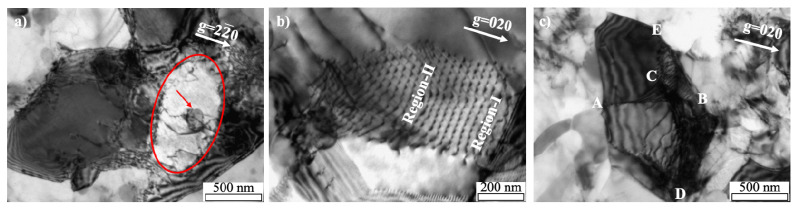
Evolution microstructure of AM5 annealed at 300 °C for (**a**) 120 s and (**b**,**c**) 600 s.

**Figure 14 materials-14-01701-f014:**
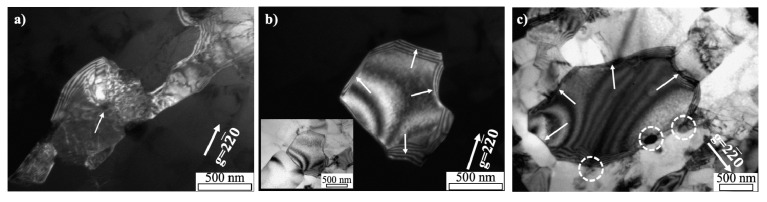
Evolution microstructure of AM9 annealed at 300 °C for (**a**,**b**) 5 s and (**c**) 60 s.

**Figure 15 materials-14-01701-f015:**
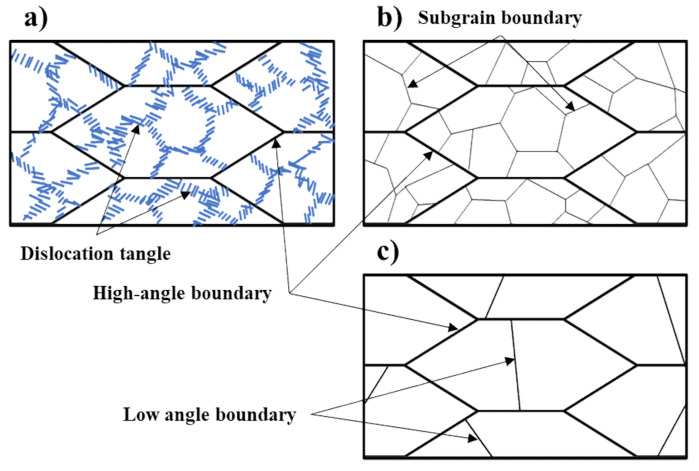
Schematic diagram for formation and coalescence of sub-grains during annealing. (**a**) cold-rolled structure with low plastic deformation, (**b**) formation of subgrain during recovery and (**c**) coalescence of the subgrain by subsequent annealing.

**Figure 16 materials-14-01701-f016:**
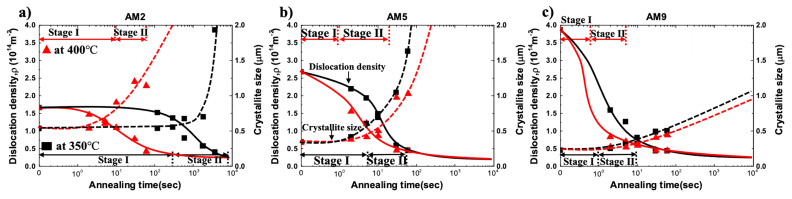
Evolution of the dislocation density and crystallite size as a function of annealing time at 350 °C and 400 °C in (**a**) AM2, (**b**) AM5, and (**c**) AM9.

**Figure 17 materials-14-01701-f017:**
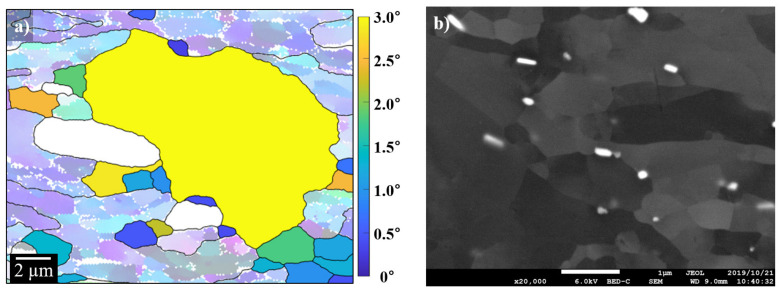
Grain orientation spread (GOS) map overlaid with IPF map in AM9 annealed at 300 °C for 30 s shows growth of recrystallized grain (**a**) in deformation zone around a coarse particle and (**b**) in dispersoid zone.

**Figure 18 materials-14-01701-f018:**
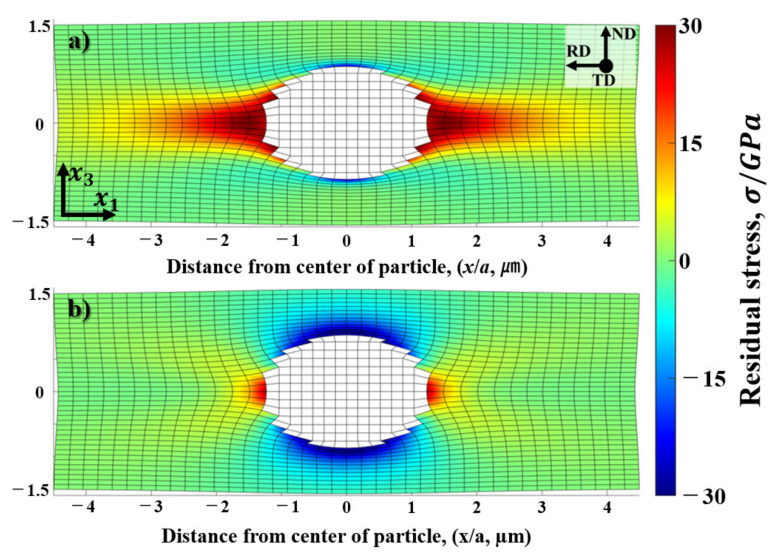
Deformed shape and residual stress around the spherical inhomogeneity (α-Al(Mn, Fe)Si precipitate) under cold-rolling condition ($\Delta \varepsilon_{33}^{p}$ = $-\Delta \varepsilon_{11}^{p}$ = 0.5, and $\Delta \varepsilon_{22}^{p}$ 0) are represented in (**a**) $\sigma_{11}$ and (**b**) $\sigma_{33}$. The displacement and residual stress outside the inhomogeneity are deduced from Equations (17) and (9), respectively.

**Figure 19 materials-14-01701-f019:**
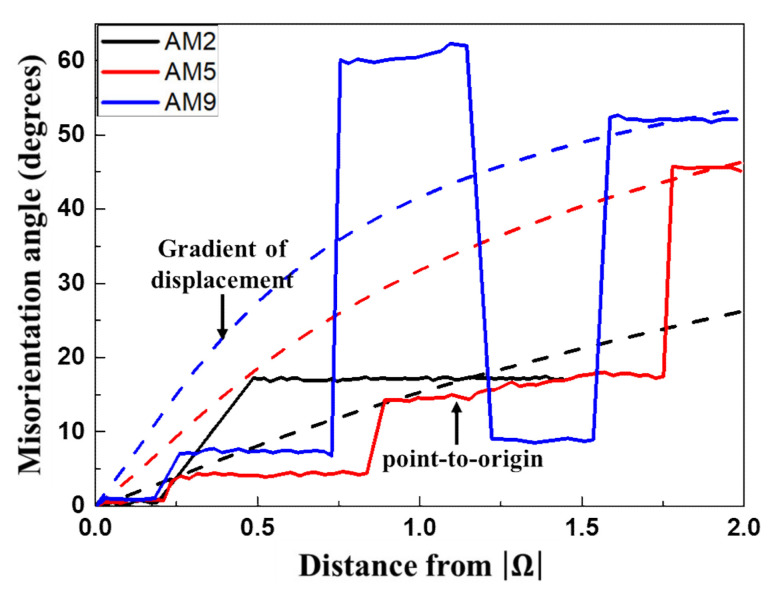
Misorientation angles profile across the rotated zone formed around the α-Al(Mn, Fe)Si precipitate measured by the prediction of displacement gradient and EBSD analysis.

**Table 1 materials-14-01701-t001:** Chemical composition of commercial AA3003 aluminum alloy used in this study (wt%).

Alloy	Mn	Cu	Fe	Si	Ti	Al
AA3003	1.15	0.16	0.52	0.28	0.01	bal.

**Table 2 materials-14-01701-t002:** Young’s modulus and Poisson ratio for aluminum matrix and α-Al(Mn, Fe)Si precipitate. The subscripts m and p indicate matrix phase and precipitate.

Constant	Value	Source
*E_m_*	69 GPa	[25]
*v_m_*	0.33	
*E_p_*	175 GPa	[26]
*v_p_*	0.28	assumed

**Table 3 materials-14-01701-t003:** Change of grain size, primary particle size, and electrical conductivity in the DC cast AA3003 aluminum alloy by homogenization.

Condition	Average Grain Size (μm)	Average Primary Particle Size (μm)	Electric Conductivity (%IACS)
**As-cast**	140	2.8	30
**As-homo**	160	4.1	43.5

**Table 4 materials-14-01701-t004:** Comparison of the residual stress inside the α-Al(Mn, Fe)Si precipitate with sphere computed by Equation (9) under cold-rolling condition ($\varepsilon_{11}^{0}$
= $-\varepsilon_{33}^{0}$ = 0.2, 0.5 and 0.9, and $\varepsilon_{22}^{0}$ = 0). Von Mises Equivalent stress, $\sigma_{\mathrm{eqv}}$, is computed by Equation (16).

$\varepsilon_{\mathrm{ij}}^{0}$	$\sigma_{\mathrm{ij}}/\mathrm{GPa}$			
	$\sigma_{11}$	$\sigma_{22}$	$\sigma_{33}$	$\sigma_{\mathrm{eqv}}$	
0.2	8.25	<10^−17^	−8.25	14.29
0.5	20.62	<10^−17^	−20.62	35.72
0.9	37.12	<10^−17^	−37.12	64.29

## Data Availability

Data sharing not applicable to this article.

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
