# Peer review of "Influence of Residual Stress around Constituent Particles on Recrystallization and Grain Growth in Al-Mn-Based Alloy during Annealing"

_materials, 2021, doi:10.3390/ma14071701_

Round 1

Reviewer 1 Report

This paper entitled "Influence of Residual Stress Around Constituent Particles on Recrystallization and Grain Growth in Al-Mn Based Alloy During Annealing” presents a detailed study on the effect of the amount of reduction and the subsequent annealing treatment on the recrystallization phenomena under the certain volume fraction of primary particle in AA 3003 alloy. This work has been competently carried out as the authors, have a well-established experience with many Al alloy and their data analysis. The manuscript could be surely published in Materials as most part of the observations is reasonably correct and well discussed, as expected from this research group. The manuscript is well-organized, and obtained results are supported by appropriate experimental data, and analyzed, the work is quite interesting. I recommend the paper be accepted for publication, after consideration of the following issues:

  1. Need to give full form so the reader is not confused about the analysis/technique being referred to in the manuscript. EBSD analysis: Electron Backscatter Diffraction (EBSD) – analysis. OIM: Orientation Imaging Microscopy. BEI: Backscattered electron image, and so on. These may be accepted abbreviations/ short forms but considering all the readers it's better to expand them for clarity purposes.
  2. English language must be improved.

Reviewer 2 Report

The study investigated the effect of residual stress on recovery and recrystallization of cold-rolled aluminum alloy AA3003. An attempt was made to solve the problem of Eshelby's inhomogeneity in cold rolling conditions. The research plan was thoroughly and reliably developed. Appropriately technically advanced research has been carried out. The microstructure and the dislocation density by TEM and X-ray sync were analyzed in detail. A number of important parameters of the aluminum alloy machining process have been established in order to obtain the appropriate product quality. These parameters are difficult to define due to the complexity of the process. The level of the English language is sufficient for the publication to be accepted. The general disadvantage of the work is the insufficiently comprehensive introduction, which could be expanded with new literature items. Due to the essence of the work, it was worth highlighting the technological value of the work, not only the scientific value. The general advantages include the wide range of research undertaken and reliability and accuracy of the obtained results.

Reviewer 3 Report

Overall, the article is well written, I got few suggestions and hope the authors can consider:

I am wondering have the authors considered using experimental techniques to determine the residual stress? Such as XRD?

MTEX code for MATLAB should be able to calculate the residual stress between grains using raw data from EBSD, have the authors considered this?

In section 2, experimental, a schematic should be given for the sampling location and orientation, with dimensions.

Regarding the definition of LAGB and HAGB, I suggest consider the article: Manufacturing a curved profile with fine grains and high strength by differential velocity sideways extrusion.

Fig.1b, Fig. 3, it would be helpful to add some brief labels in the selected zones.

Fig. 2, Fig. 5, Fig. 15, it would be better to increase the font size a bit.

The scale bar in Fig. 4a, 12, 14 is too small to be clearly seen.

Some grammar mistakes have been found:

line 52, this deformation zones

line 53, it provide

line 54, leads, ‘to’ is missing

line 87-88, The grain microstructure… were…

Reviewer 4 Report

The article is devoted to the study of the nature of recrystallization, recovery and deformation depending on the residual stresses arising around the particles of α-Al (Mn, Fe) Si in the AA3003 alloy. The article describes in detail the research methods and equipment used. The results obtained on the change in the microstructure and its characteristics during annealing and deformation are correctly presented in the form of diagrams and figures. At the same time, the authors do not provide a clear purpose of the study. However, there are some questions:

  1. Why was this alloy chosen as the object of research? Will there be similar changes in the microstructure in alloys of this system or in another alloying systems?
  2. Authors should describe in more detail the purpose of this study. It is not entirely clear for what purpose it is necessary to study the nature of recrystallization in this alloy. How can the obtained results and information be applied in practice?
  3. How was the homogenization mode of the AA3003 alloy chosen? And also it is necessary to explain the reason for choosing a temperature range of isothermal annealing: 300 ℃, 350 ℃, 375 ℃ and 400 ℃.
  4. Figure 3 shows the deformation zone around the α-Al (Mn, Fe) Si precipitate. It would be more correct to add some label for the types of these zones (to the left / right of the particle and below / above the coarse particle) near to the white and red dotted lines in the figures.

Round 2

Reviewer 3 Report

Thank you for the response and revisions. I would like to recommend it for publication.

Reviewer 4 Report

Thanks for comments, article can be accepted in present form